

# Exploring telerehabilitation awareness, application, and future outlook in sports rehabilitation among physiotherapy students: a web-based survey

Vinodhkumar Ramalingam[1,*], Jeevarathinam Thirumalai[1,*],
Ling Shing Wong[2], Rajkumar Krishnan Vasanthi[2],
Vinosh Kumar Purushothaman[2], Prathap Suganthirababu[3] and
Buvanesh Annadurai[1]

[1] Department of Sports and Manual Therapy, Saveetha College of Physiotherapy, Saveetha Institute of Medical and Technical Sciences, Chennai, Tamil Nadu, India
[2] Faculty of Health & Life Sciences, INTI International University, Nilai, Negeri Sembilan, Malaysia
[3] Department of Neurological Physiotherapy, Saveetha College of Physiotherapy, Saveetha Institute of Medical and Technical Sciences, Chennai, Tamil Nadu, India
* These authors contributed equally to this work.

Corresponding authors
Vinodhkumar Ramalingam,
vinodhkumar.scpt@saveetha.com
Ling Shing Wong,
lingshing.wong@newinti.edu.my

## ABSTRACT

Telerehabilitation is rapidly transforming the landscape of sports rehabilitation, with benefits for both recreational and elite athletes. The present study explores physiotherapy students' awareness, application, and perspectives on sports telerehabilitation. It also investigates the feasibility of incorporating telerehabilitation into physiotherapy education by examining infrastructure requirements, available resources, and student willingness. To gain insight, we conducted a global survey that yielded 524 responses from physiotherapy students. The study used a 42-item validated, self-designed questionnaire, focusing on four key aspects: screening, awareness, application, and future outlook. The results demonstrated that 65.3% of the students were familiar with the concept of telerehabilitation. However, only 19.3% had taken it as part of their formal curriculum. The most common sources of knowledge gained are social media (22.5%) and research articles (19.8%). Notably, 59.4% expressed a strong desire to include telerehabilitation in their studies, highlighting the demand for structured training. In terms of awareness, 50.3% demonstrated a moderate level, while 32.5% showed high understanding. Regarding practical implementation, 36.5% reported moderate competency, while 47.7% displayed high application knowledge. Looking ahead, 34.2% had a moderate outlook on the future of sports telerehabilitation. Meanwhile, 49.4% were highly optimistic about the role of sports telerehabilitation in physiotherapy.

## INTRODUCTION

The dynamic field of sports physiotherapy originated in the 19th century, when physiotherapy became a recognized practice (*Shaik & Shemjaz, 2014*). In a society that had celebrated athleticism, the initial practitioners understood the significance of the movements in terms of both health and performance. These practitioners laid the foundation for what would eventually evolve into the art and science of sports physiotherapy (*Huijbregts, 2010*; *Shaik & Shemjaz, 2014*). Sports physiotherapy has advanced from ancient times to the 21st century, with increased demands on athlete performance and individualized care (*Morouço, Takagi & Fernandes, 2005*). The conventional tape and the advanced modalities gave way to a new chapter of innovation with the digital platform, which utilized advanced telerehabilitation technology *via* various e-health platforms (*Brennan, Mawson & Brownsell, 2009*). Recent advancements in telehealth platforms have enhanced remote assessment, rehabilitation, and education, making telerehabilitation an essential component of modern physiotherapy training and practice (*Lee et al., 2024*).

Telerehabilitation has had a widespread impact on sports physiotherapy, affecting everyone from recreational athletes to elite professionals. It empowers athletes to push their limits and achieve new heights of performance through real-time motion analysis, AI-driven rehabilitation programs, and virtual reality-based training (*Bäcker et al., 2021*; *Lal et al., 2023*; *Seçkin, Ateş & Seçkin, 2023*; *Almansour, 2024*). These advancements have also transformed physiotherapy education, equipping students with the ability to assess and intervene remotely *via* digital health platforms and wearable sensors (*Martin et al., 2022*; *Davies et al., 2023*; *Lee et al., 2024*). Telerehabilitation offers numerous benefits, including real-time movement analysis, remote biomechanical assessment, and personalized exercise prescription under virtual supervision. This approach minimizes the risk of reinjury, contributing significantly to sports rehabilitation practices (*Baroni et al., 2023*).

Wearable technology, such as smart watches and motion sensors, plays an important role in telerehabilitation by offering continuous remote monitoring and individualized care (*LaBoone & Marques, 2024*). These devices provide real-time data, facilitating remote tracking of athletes' progress (*Kraus et al., 2021*). Sensor data, such as range of motion, joint angle, and movement speed, assist the therapist in evaluating progress (*De Fazio et al., 2023*). Additionally, wearable devices integrated with electromyography track muscle activation patterns during exercise, allowing therapists to identify compensatory movements (*Simić & Stojanović, 2024*). Beyond clinical applications, integrating training with digital assessment tools into physiotherapy education will significantly improve students' ability to conduct remote assessments and implement telerehabilitation strategies effectively, ensuring their preparedness for modern sports physiotherapy practice (*Dyck, 2019*; *Jaswal et al., 2023*; *Ramponi et al., 2024*).

Remote injury prevention strategies, such as warm-up routines and the biomechanical analysis of movement, can be implemented through wearable sensors in geographically dispersed settings (*Alzahrani & Ullah, 2024*). Recent technologies, such as physical and

biometric sensors, enable tele-assessments through haptic and augmented reality (*Annaswamy et al., 2021*; *Annadurai et al., 2023*). Telerehabilitation reduces communication barriers between athletes and trainers, allowing for a multidisciplinary approach to recovery and practice (*Muñoz-Tomás et al., 2023*; *Purushothaman et al., 2024*). Integration of these technologies into physiotherapy education ensures that future professionals develop the necessary digital competencies required to implement telerehabilitation effectively in clinical practice (*Stark-Blomeier, Krayter & Dockweiler, 2025*).

The worldwide spread of COVID-19 necessitated significant lifestyle changes and preventive measures, such as social distancing, to reduce the risk of infection transmission (*Tayech et al., 2020*). Due to the ongoing shortage of healthcare resources, exploring innovative healthcare delivery methods is crucial (*Almojaibel et al., 2020*). During the COVID-19 pandemic, measures and protocols were implemented that restricted access to outpatient rehabilitation facilities and delayed non-urgent medical procedures (*Filip et al., 2022*). These preventive measures affected outpatient services, prompting the active promotion of alternative healthcare delivery systems, such as telemedicine and e-health (*Jachak et al., 2020*; *Monaghesh & Hajizadeh, 2020*). These digital solutions were integrated not only into physiotherapy practices but also across various allied health services, ensuring that patients receive care continually despite restrictions (*Speranza et al., 2021*; *Anil et al., 2023*).

The COVID-19 global emergency accelerated the adoption of digital health solutions, including artificial intelligence (AI)-integrated telerehabilitation, which enhances communication, quality, efficiency, and accessibility to care (*Calderone et al., 2024*). These AI-powered systems analyze vast amounts of data from telerehabilitation platforms, wearable devices, and sensors to deliver precise, personalized feedback while automating key aspects of rehabilitation (*LaBoone & Marques, 2024*; *Kamalakannan et al., 2024*). One of AI's significant contributions to telerehabilitation is its ability to perform real-time motion analysis. Using advanced algorithms and computer vision, AI can assess an individual's movements during the exercises, identify deviations from correct form, and provide instant corrections (*Sitapara et al., 2023*; *Khalid et al., 2024*; *Olivas-Padilla, Manitsaris & Glushkova, 2024*). This assessment ensures that therapist-prescribed exercises are performed accurately, reducing the risk of re-injury and enhancing the effectiveness of rehabilitation (*Yang et al., 2024*). AI-based biomechanical estimation tools further allow therapists to monitor patients remotely, eliminating the need for constant in-person supervision (*Sardari et al., 2023*).

AI also facilitates multidisciplinary collaboration by integrating data from various sources, such as wearable devices, imaging, and patient records, into a centralized platform (*Schleiger et al., 2024*). This collaboration allows therapists, coaches, and healthcare providers to make informed decisions and create holistic recovery plans (*Yang et al., 2024*). Furthermore, AI's natural language processing capabilities enhance communication in telerehabilitation, bridging language barriers and improving patient education (*Sathishkumar et al., 2024*; *Schleiger et al., 2024*). Incorporating telerehabilitation into sports physiotherapy education fosters adaptability and equips students with the skills to

design and deliver remote interventions tailored to the needs of an individual athlete (*Avramescu & Monova-Zheleva, 2024*; *Ali et al., 2025*). Interactive learning modules and case-based simulations provide hands-on experience with emerging digital rehabilitation tools, ensuring that graduates are well-prepared for the evolving landscape of sports physiotherapy (*Welling & Metgud, 2023*; *Dairo, Hunter & Ishaku, 2024*).

Although technological advancements are reshaping telerehabilitation, its integration into professional practice relies on students' perspective in the field. The present study addresses a critical gap in the existing literature, predominantly focusing on general rehabilitation settings and often reporting varying levels of awareness among rehabilitation students (*Mbada et al., 2021*; *Başer Seçer & Çeliker Tosun, 2022*). However, these previous studies have primarily examined general rehabilitation, leaving a gap in understanding telerehabilitation's awareness, application, and future outlook within sports physiotherapy. The present study addresses this gap by assessing physiotherapy students' perspectives on sports telerehabilitation, identifying barriers to its implementation, and exploring its potential integration into physiotherapy education curricula to better prepare future professionals for digital rehabilitation practices.

## METHODS

### Study design and participants

We conducted a web-based cross-sectional survey targeting the physiotherapy students currently enrolled in undergraduate, postgraduate, and doctoral programs. Given the absence of a global or domestic database of physiotherapy students, the study used a non-probability sampling method to obtain preliminary insights and facilitate the recruitment of targeted physiotherapy students from diverse geographic regions. The data collection was conducted between August 2023 and March 2024. This web-based cross-sectional study adheres to the Strengthening the Reporting of Observational Studies in Epidemiology (STROBE) guideline (Supplemental File) (*STROBE Group, 2007*). Ethical approval was obtained from the Institutional Scientific Review Board of Saveetha College of Physiotherapy, Saveetha Institute of Medical and Technical Sciences (SIMATS), India (Approval No. 03/001/2023/ISRB/UGSR/SCPT). Of the 587 students who completed the survey, 524 met the selection criteria. Sixty-three students were excluded from the study as they had not completed the sports physiotherapy module at the time of the survey ($n = 42$), had duplicate responses ($n = 14$), or lacked physiotherapy background ($n = 7$). We recruited students from various regions of India, including North India (17.4%), South India (23.3%), East India (13.4%), West India (10.5%), Central India (6.9%), North East India (8.4%), and North West India (0.8%). Additionally, we included physiotherapy students from various regions outside India, such as South Asia (2.5%), Southeast Asia (10.9%), the Middle East (4%), Oceania (1.3%), and Europe (0.8%). The participants included current undergraduate students who had completed their sports physiotherapy module, postgraduate students, and students pursuing Ph.D. in Physiotherapy in various institutions within India or globally. These groups were selected to capture a range of academic exposure and practical experience in physiotherapy, contributing to a comprehensive understanding of telerehabilitation perspectives. The participants enrolled

in disciplines other than physiotherapy were excluded to maintain the study's relevance to the field and ensure focused data collection. Additionally, the participants with limited English proficiency were excluded to ensure full comprehension of telerehabilitation terms and concepts, which would enhance response accuracy and data reliability.

## Survey development and distribution

A self-constructed validated questionnaire containing 42 questions was divided into four parts: screening, awareness, application, and future outlook. A self-explanatory note accompanied the questionnaire to ensure reproducibility, allowing for consistent administration across different participant groups and study settings (Supplemental Material). The questionnaire began with an informed consent statement, requiring the participants to provide consent by clicking a checkbox before proceeding, and an expert panel validated the questionnaire contents. A pilot test was conducted with 50 participants to assess consistency and identify potential issues. Based on the findings, the questionnaire was revised, and internal consistency was confirmed using Cronbach's alpha test. A survey form was created using Google Forms and disseminated *via* several platforms, including Gmail, WhatsApp, LinkedIn, Telegram, Facebook, Line, and Instagram. It was also displayed as a QR code at conferences. In accordance with regulations set by the Institutional Scientific Review Board, we did not collect IP addresses or use cookies to maintain the anonymity of respondents.

## STATISTICAL ANALYSIS

The data were analyzed using IBM SPSS statistical software version 22.0 (IBM Corp., Armonk, NY, USA). To ensure data quality, four researchers independently performed data validation checks before the statistical analyses. Internal consistency was assessed using the Cronbach's alpha test to evaluate the reliability of the awareness, application, and future outlook sections. Descriptive statistics were used to ensure the frequency and percentage distribution of demographic, awareness, application, and future outlook sections. The chi-square test was used to analyze the associations between awareness, application, and future outlook (dependent variables) and demographic factors (independent variables). It helps to determine whether a significant association existed between demographic factors and the varying levels of telerehabilitation awareness, application, and future outlook. Ordinal logistic regression was performed to predict the likelihood of students having higher or lower awareness and application knowledge based on demographic factors, estimating odds ratios (OR). Multinomial logistic regression was used to analyze the future outlook part, which was categorized into three levels (low, moderate, and high). It allowed for a more detailed comparison of how demographic characteristics influenced students' placement in different quartile levels of future outlook, compared to treating it as a single variable. This model predicted the probability of students falling into each future outlook category based on their demographic characteristics, offering deeper insights into varying perspectives on the future of telerehabilitation. Statistical significance is defined as a *p*-value below 0.05 for a significant association.

## RESULTS

### Reliability test

Cronbach's alpha was used to test the reliability of three parts of the questionnaire. Cronbach's alpha was 0.857 for the awareness part, 0.890 for the application part, and 0.962 for the future outlook. These Cronbach's alpha values indicate acceptable reliability for the respective constructs.

### Demographic characteristics

Table 1 shows the demographic characteristics of the students. The majority, 328 (62.6%) students were in the 21–25 age range. Among the students, 286 (54.6%) were female, while 238 (45.5%) were male. The majority of students, 389 (74.2%), were from the undergraduate category, and 422 (80.5%) were from various regions across India.

### Screening questions about telerehabilitation

Screening questions were necessary to filter out students based on their knowledge about telerehabilitation (Table 2). The students with minimal prior knowledge of telerehabilitation, but who had completed the sports module in their physiotherapy curriculum, were included in the survey. Their level of awareness, application, and future outlook are summarized in Table 3.

Although 65.3% of the students were aware of the term 'telerehabilitation' and had some knowledge of e-health software, only 19.3% reported having telerehabilitation content in their physiotherapy curriculum. More than 59.4% of the students expressed interest in incorporating telerehabilitation into their curriculum. The students reported having limited exposure to telerehabilitation (23.5%), limited interest (5.7%), and an insufficient educational background (4.8%), primarily due to a lack of awareness. The remaining students were not interested, citing preferences for conventional in-person methods, lack of time or resources, or perceived irrelevance to their career goals.

### Awareness of telerehabilitation in sports physiotherapy

The maximum score for awareness was 55 (11 items). The quartiles used in the awareness part are as follows: Quartile 1 (0% to 50%), Quartile 2 (51% to 75%), and Quartile 3 (76% to 100%). Awareness levels were categorized as follows: 0 to 27.5 represents a low level of awareness, 27.51 to 41.25 represents a moderate level, and 41.26 to 55 represents a high level of awareness. A notable number of students were aware of sports telerehabilitation, with 50.3% demonstrating a moderate level of awareness and 32.5% showing a high level of awareness (Table 3).

In Table S1 48.5% of the students agreed that they could be able to differentiate between real-time and store-and-forward telerehabilitation approaches, while a significantly larger proportion (61.1%) of the students could distinguish between telerehabilitation and telemedicine, demonstrating understanding of their unique uses. Awareness of common sports injuries amenable to telerehabilitation was relatively high, with 55.8% of the students agreeing that such conditions are suitable for remote management. Additionally,

**Table 1 Summary of students demographic including age, gender, academic level and region.**

| Demographics | Frequency (n) | Percentage (%) |
|---|---|---|
| **Age** | | |
| 18–20 | 143 | 27.3% |
| 21–25 | 328 | 62.6% |
| >25 | 53 | 10.1% |
| **Gender** | | |
| Male | 238 | 45.5% |
| Female | 286 | 54.6% |
| **Academic level** | | |
| UG* | 389 | 74.2% |
| PG* | 114 | 21.8% |
| Ph.D.* | 21 | 4.0% |
| **Region** | | |
| Domestic realm (India) | 422 | 80.5% |
| Global realm | 102 | 19.5% |

Note:
*UG, Undergraduate; PG, Postgraduate; Ph.D. Doctorate of Philosophy.

**Table 2 Screening questions about telerehabilitation among physiotherapy students.**

| Survey questions | Frequency | Percentage |
|---|---|---|
| Are you familiar with the term "telerehabilitation" and its applications, such as video calls, mobile apps, and software tools for rehabilitation purposes? | | |
| Yes | 342 | 65.3% |
| No | 182 | 34.7% |
| If yes, where did you first come across the term "telerehabilitation"? | | |
| Academic courses | 73 | 13.9% |
| Research articles | 104 | 19.8% |
| Social media | 118 | 22.5% |
| Workshops or seminars | 47 | 9.0% |
| If not, why are you not familiar with the term "telerehabilitation" | | |
| Lack of exposure/information | 123 | 23.5% |
| Limited interest in healthcare topics | 30 | 5.7% |
| Insufficient educational background | 25 | 4.8% |
| Others | 04 | 0.8% |
| Have you received any formal education on telerehabilitation as part of your curriculum? | | |
| Yes | 101 | 19.3% |
| No | 241 | 46.0% |
| Would you be interested in receiving education or training about telerehabilitation techniques as part of your physiotherapy curriculum? | | |
| Yes | 311 | 59.4% |
| No | 31 | 5.9% |
| If "yes", what specific aspects of telerehabilitation education would you find most valuable? | | |
| Practical application of telerehabilitation techniques | 144 | 27.5% |

(Continued)

| Survey questions | Frequency | Percentage |
|---|---|---|
| Understanding telecommunication technologies | 34 | 6.5% |
| Patient assessment and monitoring through telehealth | 121 | 23.1% |
| Legal and ethical considerations | 12 | 2.3% |
| If "no" what factors contribute to your lack of interest in receiving telerehabilitation education as part of your sports study's curriculum? | | |
| Already knowledgeable on the subject | 04 | 0.8% |
| Prefer traditional in-person instruction | 17 | 3.2% |
| Not relevant to my career goals | 05 | 1.0% |
| Lack of time or resources | 05 | 1.0% |

**Table 3 Frequency distribution of awareness, application knowledge, and future outlook levels among physiotherapy students based on tertile categorization.** The distribution of awareness, application knowledge, and future outlook levels was based on quartile categorization. The maximum possible scores were 55 for awareness (11 items), 65 for application knowledge (13 items), and 55 for future outlook (11 items). Quartile classification was applied as follows: Quartile 1 (0–50%), Quartile 2 (51–75%), and Quartile 3 (76–100%). Awareness levels were categorized as low (0–27.5), moderate (27.51–41.25), and high (41.26–55). Application knowledge levels were classified as low (0–32.5), moderate (32.51–48.75), and high (48.76–65). Future outlook levels were defined as low (0–27.5), moderate (27.51–41.25), and high (41.26–55).

| Variable | Frequency | Percentage |
|---|---|---|
| **Awareness** | | |
| Low | 59 | 17.3% |
| Moderate | 172 | 50.3% |
| High | 111 | 32.5% |
| **Application knowledge** | | |
| Low | 54 | 15.8% |
| Moderate | 125 | 36.5% |
| High | 163 | 47.7% |
| **Future outlook scope** | | |
| Low | 56 | 16.4% |
| Moderate | 117 | 34.2% |
| High | 169 | 49.4% |

59.6% recognized the potential role of telerehabilitation in supporting post-surgical recovery among athletes.

Notably, awareness of professional certification courses related to telerehabilitation remained low, with only 43% of the students indicating agreement. Perceptions regarding the availability of adequate support and resources for students interested in telerehabilitation were divided, as 39.2% agreed while 41.2% disagreed. Regarding the acceptance of telerehabilitation, 65.8% of the students believed that practical physiotherapy techniques are essential for effective sports rehabilitation, suggesting that telerehabilitation may be less suitable. However, 66.3% of the students indicated that

incorporating telerehabilitation into the education curriculum would enhance their interest and willingness to use it for sports rehabilitation. Concerning the research and innovations in telerehabilitation, 45.9% of the students were aware of those developments. Moreover, 54.7% viewed telerehabilitation as an environment-friendly replacement for conventional in-person methods. A moderate proportion (43.9%) of the students acknowledged awareness of ethical considerations related to sports telerehabilitation.

Table S2 shows the association between awareness of telerehabilitation and the demographic characteristics of the students, which was analyzed using the chi-square test. Age was significantly associated with awareness of telerehabilitation ($\chi^2$ = 27.541, $p$ = 0.001) with a very strong association ($\varphi$ = 0.284). Gender ($\chi^2$ = 8.810, $p$ = 0.08) and the academic level ($\chi^2$ = 13.798, $p$ = 0.08) of the students did not establish any significant association with awareness of telerehabilitation. Region was significantly associated with awareness of telerehabilitation ($\chi^2$ = 11.665, $p$ = 0.02), indicating a strong association ($\varphi$ = 0.185).

The impact of demographic variables on sports telerehabilitation awareness was analyzed using ordinal logistic regression, as this method assesses how demographic factors influence the likelihood of awareness (Table S3). The regression coefficient for female students was −0.29 ($p$ = 0.152), corresponding to an OR of 0.748 (95% CI [0.503–1.113]). This finding suggests that female students were about 25% less likely to report higher awareness compared to male students; however, this difference was not statistically significant. Age was analyzed in three categories, with students over 25 years as the reference group. The students aged 18–20 years had a regression coefficient of 0.472 ($p$ = 0.260), with an OR of 1.60 (95% CI [0.705–3.642]), indicating they were about 1.6 times more likely to report higher awareness compared with the reference group, though this difference was not statistically significant. Similarly, the students aged 21–25 years had a coefficient of −0.210 ($p$ = 0.570), with an OR of 0.811 (95% CI [0.393–1.673]), suggesting they were about 19% less likely to report higher awareness, which also was not statistically significant.

The impact of academic level and region on the students' awareness of sports telerehabilitation was also analyzed using ordinal logistic regression. When compared to the Ph.D. students (reference group), the undergraduate students had a regression coefficient of −0.173 ($p$ = 0.739), with an OR of 0.841 (95% CI [0.303–2.332]), suggesting they were about 16% less likely to report higher awareness; however, the result was not statistically significant. The postgraduate students had a regression coefficient of −0.403, corresponding to an OR of 0.668 (95% CI [0.240–1.857]), indicating they were about 33% less likely to report higher awareness relative to the Ph.D. students, but this association was also not statistically significant.

By contrast, region was a statistically significant predictor. The students residing in India had a regression coefficient of 0.669 ($p$ = 0.013), with an OR of 1.952 (95% CI [1.153–3.308]), indicating they were nearly twice as likely to have higher awareness compared to students from other countries. This significant regional difference could be attributed to the greater exposure to telerehabilitation practices or varying educational emphases in physiotherapy programs. These findings highlight the need for tailored

educational strategies aimed at enhancing telerehabilitation awareness, particularly among students at lower academic levels and in regions with limited exposure.

## Application knowledge on telerehabilitation in sports physiotherapy

The maximum score for application knowledge was 65 (13 items). The quartiles used for categorization are as follows: Quartile 1 (0–50%), Quartile 2 (51–75%), and Quartile 3 (76–100%). Application knowledge levels are classified as follows: 0–32.5 represents a low level, 32.51–48.75 represents a moderate level, and 48.76–65 represents a high level. Table 3 presents the distribution of application knowledge levels. A notable number of the students demonstrated application knowledge of telerehabilitation in the sports realm, with 36.5% reporting a moderate level and 47.7% reporting a high level of application knowledge (Table 3).

Table S4 shows that 55.6% of the students believed that offering telerehabilitation as a treatment choice reduces the overall cost of sports rehabilitation for athletes and organizations. Despite that, 44.4% of the students indicated some uncertainty about its cost-effectiveness. When considering the availability of resources, including software and technologies, 58.4% agreed that resource availability affects the effectiveness of telerehabilitation. Other students highlighted their concerns about resource limitations. A majority of the students (76%) agreed that telerehabilitation can bridge geographical gaps, establishing access to high-quality physiotherapy services in remote areas, while 78.9% agreed that telerehabilitation ensures effective communication and motivation for athletes.

The integration of telerehabilitation with virtual reality to create engaging rehabilitation experiences was supported by 65.8% of the students, while some expressed disbelief in this technological advancement. Telerehabilitation achieving the same outcomes as conventional in-person rehabilitation is observed heterogeneously, with 36.2% agreeing and 43% disagreeing. This heterogeneity reveals that the students see the potential aid of telerehabilitation, while others are skeptical of its effectiveness compared to conventional methods. Approximately 55% of the students believed that athletes favor quicker access to telerehabilitation services, indicating a need for further upgrades to enhance its accessibility. A significant number of the students (73.4%) believed that telerehabilitation contributes to a sustainable approach by reducing the need for travel resources.

The statement that telerehabilitation enhances access to sports rehabilitation expertise for amateur athletes and those with economic problems was supported by 54.6% of the students. Approximately 64% of the students agreed on the necessity of ongoing research and evidence-based practice for the implementation of telerehabilitation. Most students (58.5%) favored the development of a specialty course for telerehabilitation within the physiotherapy discipline. However, 27.2% of the students opposed the inclusion of such specialty courses, while 69.6% supported the capability of telerehabilitation to effectively monitor athlete progress, reflecting strong confidence in its tracking potential. Additionally, 55.9% believed that telerehabilitation contributes to reducing recovery time following sports injuries. However, 28.6% disagreed with the previous statement, highlighting diverse perspectives regarding telerehabilitation effectiveness in accelerating rehabilitation.

Table S5 shows the association between the students' application knowledge of telerehabilitation and the demographic characteristics, which was analyzed using the chi-square test. A significant association was found between age and the application knowledge of telerehabilitation ($\chi^2$ = 35.215, $p < 0.0001$), with a strength of association ($\varphi$ = 0.227). Additionally, gender demonstrated a significant association with the application of telerehabilitation ($\chi^2$ = 12.272, $p$ = 0.015), with a strength of association ($\varphi$ = 0.189). The association between students' academic level and the application of telerehabilitation was not statistically significant ($\chi^2$ = 13.603, $p$ = 0.093). Also, region had no significant association with the application of telerehabilitation ($\chi^2$ = 6.972, $p$ = 0.137).

The impact of demographic predictors on application knowledge of sports telerehabilitation was examined using ordinal logistic regression, as detailed in Table S6. Gender was not a significant predictor, with a regression coefficient of −0.228 for females ($p$ = 0.261), corresponding to an OR of 0.796. This finding suggests that female students were about 20% less likely to demonstrate higher levels of application knowledge compared to males; however, this difference was not statistically significant. Age was found to be a significant predictor. Students aged 18–20 had a regression coefficient of −1.509 ($p < 0.001$), OR = 0.221, with a 95% confidence interval (CI) of [0.513–0.950], indicating they were approximately 78% less likely to report higher application knowledge compared with those over 25 years. Conversely, the 21–25 age group had a coefficient of −0.339 ($p$ = 0.367), OR = 0.712, suggesting they were about 29% less likely to report higher application knowledge, though this difference was not statistically significant.

Academic level did not significantly predict application knowledge. The undergraduate students had a coefficient of −0.149 ($p$ = 0.779, OR = 0.862), suggesting they were about 14% less likely to demonstrate higher application knowledge than the Ph.D. scholars. The postgraduate students had a coefficient of −0.522 ($p$ = 0.324, OR = 0.593), indicating they were about 41% less likely to do so. However, neither difference was statistically significant. Region also did not have a statistically significant effect. The students from India had a regression coefficient of −0.305 ($p$ = 0.259), OR = 0.737, suggesting they were about 26% less likely to report higher application knowledge than the students from other countries, though this difference was not statistically meaningful.

These findings highlight age, particularly the 18 to 20 age group, as the most influential demographic factor associated with lower levels of knowledge regarding the application of sports telerehabilitation. This finding underscores the need to integrate telerehabilitation training earlier in physiotherapy education to strengthen the digital rehabilitation competencies of younger students.

## Future outlook on telerehabilitation in sports physiotherapy

The maximum score for future outlook was 55 (11 items). The quartiles used for categorization were as follows: Quartile 1 (0–50%), Quartile 2 (51–75%), and Quartile 3 (76–100%). Future outlook levels were classified as follows: 0–27.5 represents a low level, 27.51–41.25 represents a moderate level, and 41.26–55 represents a high level. The distribution of future outlook levels is presented in Table 3. A notable number of students have a positive future outlook on telerehabilitation in the sports realm, with 34.2% showing
a moderate outlook and 49.4% demonstrating a high level of application knowledge (Table 3).

In Table S6, 56.4% of the students expressed a willingness to invest time and effort in staying updated on advancements in telerehabilitation technologies. While 45.1% supported the feasibility of telerehabilitation as a standard practice in sports rehabilitation, 54.9% remained neutral or expressed doubt, indicating uncertainty regarding its practical implementation. A majority (57.6%) considered the integration of telerehabilitation into the physiotherapy curriculum to be essential, though 27.2% were neutral and 15.2% disagreed, reflecting mixed opinions. A belief in the growing acceptance and recognition of telerehabilitation within the healthcare sector was held by 60.5% of the students, while 39.4% expressed hesitation or disagreement. Furthermore, 60.8% anticipated that telerehabilitation would evolve to address challenges in sports rehabilitation, although 12.6% were uncertain and 26.6% disagreed. Most students perceived a benefit in utilizing telerehabilitation for personalized AI-guided routines, with 57.9% expressing agreement. Meanwhile, 26.6% remained neutral, and 15.5% disagreed. Regarding the future potential of telerehabilitation for neurofeedback to enhance mental resilience during recovery, 45.9% supported the idea, while 40.6% were neutral and 13.5% expressed disagreement, reflecting diverse perspectives.

Nearly half of the students (44.4%) acknowledged the influence of regulatory or legal aspects on preparedness to incorporate telerehabilitation, while 43.3% remained neutral, indicating notable uncertainty regarding its impact. Of the students, 50.3% perceived telerehabilitation as an effective means of enhancing athlete engagement and promoting active participation in rehabilitation, while 35.1% remained neutral and 14.6% disagreed, indicating varied perceptions among participants. Technical issues, such as poor network connectivity, were seen as a limiting factor by 61.1% of the students, whereas 24.3% were neutral and 14.7% did not view it as a significant barrier. Furthermore, 59.6% identified difficulties in accurately assessing athlete progress through telerehabilitation, though 26.9% were neutral and 13.5% did not perceive this as a concern.

Table S7 shows the association between the future outlook of telerehabilitation and the demographic characteristics of students, which was analyzed using the chi-square test. Age was significantly associated with the future outlook of telerehabilitation ($\chi^2$ = 31.621, $p$ < 0.0001), showing a weak association ($\varphi$ = 0.215). Gender did not show a significant association ($\chi^2$ = 6.831, $p$ = 0.141). The association between students' academic level and the future outlook of telerehabilitation was not statistically significant ($\chi^2$ = 9.972, $p$ = 0.267). Also, the region showed no significant association with the future outlook of telerehabilitation ($\chi^2$ = 8.131, $p$ = 0.087).

Table S8 presents the results of a multinomial logistic regression analysis examining demographic predictors of future outlook toward telerehabilitation, with the moderate category used as the reference. The analysis used a quartile-based categorization to assess how various demographic factors influenced participants' perceptions. The intercept for the low-level future outlook category was not statistically significant ($p$ = 0.365), indicating that, in the absence of demographic influences, the likelihood of having a low future outlook does not significantly differ from the moderate category.

Gender emerged as a statistically significant predictor. The regression coefficient for females was –0.866 (Wald = 6.099, $p$ = 0.014), with an OR of 0.420. This finding indicates that, compared with males, female students were significantly 58% less likely to report a low future outlook on telerehabilitation. In other words, female students showed a more optimistic perspective toward the future of telerehabilitation.

The regression analysis further identified age as a significant predictor of future outlook toward telerehabilitation. For the 18–20 age group, the regression coefficient was 3.423 (Wald = 7.269, $p$ = 0.007), with an OR of 30.676. This finding means that individuals aged 18–20 were over 30 times more likely to report a low future outlook compared to those over 25 years, indicating a significantly more pessimistic perspective. Similarly, for the 21–25 age group, the coefficient was 2.697 ($p$ = 0.023), with an OR of 14.840, meaning they were nearly 15 times more likely to report a low future outlook than those aged over 25, thereby reflecting a significantly less positive perception. Regarding academic level, the undergraduate students had a regression coefficient of –1.255 ($p$ = 0.334), with an OR of 0.285, suggesting they were about 71% less likely to report a low outlook than the Ph.D. students, although this result was not statistically significant. The postgraduate students also showed no significant difference (coefficient = 0.109, $p$ = 0.931), indicating outlook levels similar to the Ph.D. scholars. Region emerged as a significant predictor. The students from India had a regression coefficient of –1.424 ($p$ = 0.002), with an OR of 0.241, indicating they were about 76% less likely to report a low future outlook than the students from other regions. This finding suggests a more positive perception of telerehabilitation among the Indian students.

The intercept for the high level of future outlook category was not statistically significant ($p$ = 0.816), suggesting that, without considering demographic factors, the chances of having a high future outlook are not significantly different from having a moderate one. Gender was a significant predictor, with a regression coefficient of –0.527 ($p$ = 0.047) and an OR of 0.590. This data means that female students were about 41% less likely to report a high future outlook compared to male students. Age also played a significant role. The students aged 18–20 had a coefficient of –1.810 ($p$ = 0.001), with an OR of 0.164, meaning they were about 84% less likely to express a high future outlook compared to those over 25 years. By contrast, the 21–25 age group did not show a significant difference (coefficient = –0.263, $p$ = 0.579, OR = 0.769), indicating similar future outlook levels to the older age group.

Academic level analysis indicated that the undergraduate students had a regression coefficient of 1.056 ($p$ = 0.099), which suggests a potential but not statistically significant association. The OR of 2.875 implies that undergraduates were nearly three times more likely to report a high future outlook than the Ph.D. students, though this result should be interpreted with caution. The postgraduate students also showed no significant association (coefficient = 0.524, $p$ = 0.407, OR = 1.689), indicating only a slight, non-significant increase in the likelihood of a high future outlook. Region was a statistically significant predictor. The students from India had a regression coefficient of 0.733 ($p$ = 0.046) and an OR of 2.082, meaning they were more than twice as likely to report a high future outlook toward telerehabilitation compared to students from other regions. This finding highlights

notable regional differences in optimism about the future use of telerehabilitation technologies.

## DISCUSSION

This research was focused on sports telerehabilitation awareness, application, and future outlook among physiotherapy students. Our study findings highlight a positive attitude among the physiotherapy students toward telerehabilitation platforms and technologies. The students demonstrated a moderate level of awareness and a moderate to high level of application knowledge. Additionally, they expressed confidence that telerehabilitation will become a standard practice in sports rehabilitation. The responses from physiotherapy students in India and outside India were analyzed to gain a better understanding of the role of telerehabilitation in sports physiotherapy and its accessibility in universities and colleges.

### Screening of physiotherapy students about sports telerehabilitation awareness

The screening phase of the study was performed to scrutinize the baseline knowledge of physiotherapy students regarding sports telerehabilitation. A key objective of this screening part was to ensure that the students had sufficient knowledge of sports telerehabilitation to participate in the survey. A substantial number of the students were familiar with the term 'telerehabilitation,' having primarily learned about it through social media platforms, highlighting the significant role social media plays in raising awareness about telerehabilitation (*Başer Seçer & Çeliker Tosun, 2022*). The students who are not familiar with the term indicated a lack of exposure and limited educational background as the major reasons, implying the necessity of incorporating telerehabilitation education into the physiotherapy curriculum to prevent the knowledge gap (*Medoff-Cooper et al., 2015*). Most students expressed interest in receiving education on telerehabilitation in their curriculum, particularly focusing on practical patient virtual assessment and telecommunication technologies, and this interest indicates an understanding of telerehabilitation's importance in sports physiotherapy (*Başer Seçer & Çeliker Tosun, 2022*).

### Awareness

The physiotherapy students exhibited varying levels of awareness regarding the multidimensional aspects of telerehabilitation. Most demonstrated a moderate level of awareness of sports telerehabilitation, reflecting a basic understanding of its concepts. This finding highlights the need for more comprehensive education and professional development opportunities. The COVID-19 pandemic further accelerated the integration of telehealth services, emphasizing the importance of preparing future healthcare professionals to effectively utilize advanced technologies (*Wosik et al., 2020*; *Gilmutdinova et al., 2021*; *Kim et al., 2022*). Telerehabilitation is an effective approach for managing various conditions by enabling remote assessment, monitoring, and intervention in chronic disease management. It allows for continuous tracking of vital signs, medication

adherence, and lifestyle modifications, ensuring timely and appropriate interventions (*Anton et al., 2018*).

The students' levels of awareness were shaped by their prior exposure to telerehabilitation, whether through academic training or informal sources such as research articles, social media, and workshops (*Shaw, 2023*). Those with greater external exposure demonstrated higher awareness, whereas those with limited exposure showed lower understanding. This discrepancy points to the lack of standardization in physiotherapy curricula, where telerehabilitation is not consistently included, leading to uneven knowledge among the students. Moreover, limited familiarity with digital health technologies likely contributed to lower awareness scores, as the students with less exposure to healthcare innovations struggled to grasp telerehabilitation concepts.

Demographic factors were found to significantly influence the awareness levels of physiotherapy students. Notably, students in the 18–20 age group exhibited a higher level of awareness than those over 25 years old. This trend aligns with the findings of *Mbada et al. (2021)*, who described the younger generation's status as "digital natives," individuals who have grown up using smartphones, computers, and other digital platforms, naturally enhancing their comfort and familiarity with emerging technologies (*Wosik et al., 2020*). However, despite their higher awareness, the younger students demonstrated lower application proficiency, highlighting a gap between theoretical knowledge and practical implementation. This paradox may be due to curriculum limitations. While the younger students acquire theoretical knowledge through informal sources such as social media, they frequently lack structured opportunities to engage with telerehabilitation tools in clinical practice. In addition to age, regional differences also revealed a considerable impact on awareness levels. The students from technologically advanced regions or countries with strong digital infrastructure and established telehealth policies demonstrated a higher level of familiarity with telerehabilitation. By contrast, those from under-resourced or rural areas reported lower levels of awareness, often due to limited access to stable internet connectivity, fewer digital learning platforms, and minimal exposure to telehealth practices during their academic training. The disparity underscores the importance of equitable access to digital resources across institutions, which remains a challenge in many developing countries. These variations further emphasize the critical role of institutional policies and national-level health education strategies. Universities that actively integrate e-health education into their physiotherapy programs tend to produce students who are not only more aware but also more competent in digital health practices. By contrast, lack of institutional emphasis on telehealth reduces opportunities for students to develop essential skills, regardless of their age or interest.

Notably, gender and academic level did not show significant associations with awareness, suggesting that telerehabilitation knowledge is not inherently linked to these variables. Instead, telerehabilitation knowledge is shaped more by individual access to technology, exposure to digital learning environments, and the structure of the educational program itself (*Bharath, Sinduja & Lakshmi, 2021*; *Mbada et al., 2021*). The increasing incorporation of digital tools within physiotherapy education, such as online lectures, virtual simulations, and telehealth modules, has contributed to a more equitable

distribution of digital health knowledge across genders. As digital literacy becomes more widespread, gender-based differences in familiarity with telerehabilitation have diminished, leading to a more uniform understanding of digital health technologies (*Nuara et al., 2022*).

Beyond demographic trends, the survey reinforces the practical relevance of sports telerehabilitation, especially in post-surgical rehabilitation and the management of common musculoskeletal injuries. It emerges as a viable complement or even an alternative to conventional in-person physiotherapy. Existing literature, including a systematic review and meta-analysis (*Agostini et al., 2015*), highlight telerehabilitation's ability to deliver cost-effective, accessible, and efficient care without compromising treatment outcomes. It offers particular advantages in addressing geographical limitations, enabling remote patient engagement while maintaining comparable improvements in function and recovery (*Agostini et al., 2015*; *Nuara et al., 2022*). Given these benefits, integrating e-health platforms into physiotherapy education becomes not just beneficial but essential. The World Health Organization (WHO) recognizes healthcare professionals as key drivers in the adoption and success of e-health solutions, particularly as health systems grow more technologically complex (*World Health Organization & Union IT, 2020*; *Manyazewal et al., 2021*). The findings from this study also indicate that the students exposed to e-health training are more likely to advocate for its application in clinical settings. Equipping future physiotherapists with digital competencies will enhance their readiness to utilize telerehabilitation and contribute to its broader acceptance in routine clinical practice (*Holland Brown & Bewick, 2023*).

However, it is important to acknowledge the influence of potential confounding variables that may have affected the awareness outcomes. One key factor is prior exposure to telerehabilitation, which can vary significantly depending on students' curriculum, personal interest, and access to external learning resources such as academic publications, online forums, and professional workshops. This uneven exposure likely influenced both their awareness and self-perceived application skills. In some cases, the students may lack understanding simply due to insufficient curriculum coverage, minimal clinical exposure, or a lack of engagement with digital health topics. These discrepancies highlight the pressing need for standardized, inclusive training in telerehabilitation across institutions to ensure that all students, regardless of background, are equally prepared for the evolving demands of digital physiotherapy.

### Application

This study observed positive application knowledge among the physiotherapy students, reinforcing the growing acceptance of telerehabilitation in clinical practice. Telerehabilitation's increasing adoption aligns with broader healthcare trends emphasizing the role of e-health technologies (*Saaei & Klappa, 2021*; *Purushothaman et al., 2024*). However, the students aged 18–20 years exhibited lower application proficiency than older students, emphasizing the importance of clinical experience in skill development (*Anton et al., 2018*). Remarkably, demographic factors such as gender, academic level, and region did not show significant predictive value. These findings align with previous findings

suggesting that individual experiences, academic exposure, institutional support, and professional aspirations play a more substantial role in shaping student perspectives (*Brennan, Mawson & Brownsell, 2009*).

Students' proficiency in applying telerehabilitation appears to be strongly influenced by hands-on experience and academic training. Those with prior exposure to telerehabilitation through coursework, internships, or independent learning were more likely to demonstrate confidence in implementing telerehabilitation strategies. By contrast, the students with limited or no practical experience struggled to conceptualize real-world applications, resulting in lower competency levels. Access to technological resources also emerged as a critical factor. The students with experience using digital platforms for assessments, consultations, or rehabilitation planning reported greater competence, whereas those unfamiliar with telehealth tools found it difficult to envision its practical integration into physiotherapy.

A significant number of the students acknowledged that telerehabilitation could reduce healthcare costs while expanding service delivery across various domains such as sports rehabilitation, chronic disease management, mental health, and elderly care. This finding highlights telerehabilitation's potential to enhance accessibility and reduce financial burdens for both institutions and patients, especially in resource-limited settings (*Snoswell et al., 2020*). It also bridges geographical gaps, proving invaluable during critical times like the COVID-19 pandemic, and improves access to care in remote or underserved regions (*Snoswell et al., 2020*; *Gajarawala & Pelkowski, 2021*). Nevertheless, limitations in technological infrastructure and internet reliability remain pressing challenges, echoing barriers identified in global literature (*Mumtaz et al., 2023*).

The integration of advanced technologies, such as virtual reality, has the potential to further enhance telerehabilitation by increasing engagement, interactivity, and patient motivation, ultimately improving adherence and clinical outcomes (*Schröder et al., 2019*). Moreover, telerehabilitation can facilitate continuous monitoring and real-time feedback, which are essential for monitoring the patient's progress (*Patel et al., 2021*). Despite these advantages, effective implementation still faces multiple challenges. Reliable internet access and the availability of suitable devices remain significant concerns (*Mumtaz et al., 2023*). Telerehabilitation demands a robust digital infrastructure to support efficient communication between patients and healthcare providers (*Brennan, Mawson & Brownsell, 2009*). Moreover, the lack of formal training limits its widespread use. Structured educational programs covering technical competencies, remote communication strategies, and best practices for telerehabilitation are essential. Integrating these components into the curriculum, along with practical hands-on training, will better equip future physiotherapists to adopt and apply telerehabilitation effectively in their professional practice.

## Future outlook

The physiotherapy students demonstrated a strong willingness to adopt telerehabilitation, reflecting broader advancements in digital healthcare and the increasing role of technology-driven rehabilitation (*Phuong et al., 2023*). They also expressed readiness to

invest time and effort to stay updated on recent advancements in telerehabilitation technology. This data reflects the growing necessity for healthcare professionals to continually upgrade their skills and stay informed about technological innovations.

Gender showed mixed significance, with females less likely to report both low and high future outlooks, suggesting a more neutral stance. These findings align with those of literature suggesting that gender in telerehabilitation often adopts a cautious stance toward new technologies due to concerns about practical feasibility (*Veras et al., 2025*). Meanwhile, regional differences were a significant predictor of future outlook, potentially due to variations in policy support, professional opportunities, and healthcare infrastructure (*Peretti et al., 2017*). Access to training programs and the adoption of digital health practices may also contribute to regional disparities in how students perceive the future of telerehabilitation (*Shaw, 2023*). Age also played a role in shaping students' perspectives. The students aged 18–20 and 21–25 were less likely to exhibit a high future outlook compared to students over 25, indicating that increased professional experience and exposure to telerehabilitation pathways positively influence confidence in its future application (*Stark-Blomeier, Krayter & Dockweiler, 2025*). This finding supports earlier findings that profession-related experience deepens understanding and acceptance of telerehabilitation (*Başer Seçer & Çeliker Tosun, 2022*).

Career aspirations and interest in digital healthcare further influenced students' future outlooks. Those aiming to work in technology-driven healthcare settings or research environments tended to show more optimism. Conversely, the students favoring conventional physiotherapy approaches appeared more hesitant about integrating telerehabilitation into future practice (*Shaw, 2023*). A lack of formal curriculum integration may have negatively impacted perceptions of telerehabilitation's long-term relevance. Limited exposure to digital health content and emerging trends may lead students to underestimate the role of telerehabilitation in physiotherapy education. Structured curriculum integration would help enhance competence and build confidence in future applications (*Coetzee et al., 2022*).

Despite a generally positive outlook, the students identified several challenges that must be addressed for telerehabilitation to achieve its full potential. Common concerns included network instability, limited access to high-speed internet, and regulatory issues. In regions with weak digital infrastructure, connectivity issues hinder real-time communication and data transfer, reducing the effectiveness of remote interventions. Accurate monitoring of patient progress also emerged as a concern, emphasizing the need for standardized tools to improve remote rehabilitation quality (*Bouabida, Lebouché & Pomey, 2022*). Many students have reported that telerehabilitation will become a standard practice in sports rehabilitation, supported by growing e-health acceptance across healthcare sectors (*Hossain et al., 2019*). Studies have predicted continued expansion of telerehabilitation, particularly in overcoming access barriers in underserved regions (*Mbada et al., 2021*; *Coetzee et al., 2022*). The integration of advanced technologies such as AI-guided exercise routines and neurofeedback could further enhance rehabilitation outcomes (*Afyouni, Einea & Murad, 2019*; *Le, Loomes & Loureiro, 2023*). In sports settings, telerehabilitation

encourages athletes to take active roles in their recovery, highlighting how technology supports and optimizes rehabilitation progress (*Tenforde et al., 2020*).

Nonetheless, the students emphasized that accurate assessment of athlete progress remains a significant challenge. Effective implementation requires enhanced technological infrastructure and dedicated tools for tracking outcomes (*Dunphy & Gardner, 2020*; *Leochico et al., 2020*; *Gardner, Podbielski & Dunphy, 2024*). Ongoing research and development are essential to fully realize the benefits of telerehabilitation. Addressing technical gaps, developing clear policies, and standardizing curriculum content are critical for successful integration into clinical practice (*Terrell et al., 2021*; *Skempes et al., 2022*). Policymakers must also resolve key issues such as reimbursement models, data privacy, and telehealth standardization (*Watzlaf et al., 2017*; *Abernethy et al., 2022*). Transparent regulatory frameworks will be important to foster trust among healthcare professionals and support the integration of telerehabilitation into physiotherapy education and clinical care (*Kim et al., 2022*; *Al Meslamani, 2023*).

## CONCLUSION

This study emphasizes a mixed outlook among physiotherapy students towards adopting telerehabilitation technologies. The students demonstrated a moderate level of awareness and application knowledge towards telerehabilitation which informs a variation in their confidence and readiness to incorporate it in their future practice. Notably, the younger students demonstrated higher awareness but lower application proficiency, likely due to their greater digital literacy, though they may lack the practical experience required for effective application, as supported by the digital literacy framework.

Despite these gaps, the students are confident that telerehabilitation will become standard practice in sports rehabilitation, and they are willing to invest time in staying updated with trending technologies, which reveals a high level of perception towards the future outlook. To improve students' awareness and application knowledge, it is important to incorporate hands-on training with telerehabilitation tools into the curriculum for the students in utilizing these technologies. Additionally, collaboration between institutions and technology experts in telerehabilitation can enhance the adaptability in real-time practice.

Given the identified gaps in both awareness and application, educational reforms are essential to equip future physiotherapists with the necessary skills and knowledge. These reforms should focus on integrating telerehabilitation into physiotherapy curricula through dedicated coursework, hands-on training, and interdisciplinary collaborations. Additionally, incorporating virtual simulation-based learning and standardized competency assessments will enhance practical exposure. Policy initiatives should support faculty development programs, funding for telerehabilitation research, and collaborations with healthcare technology industries to ensure students are well-versed in real-world applications. By addressing these gaps, educational institutions can foster innovation, improve accessibility, and enhance the quality of physiotherapy education, ultimately enabling graduates to deliver higher-quality healthcare to athletes and the broader population.

## LIMITATIONS

Students self-selected into the study, introducing a risk of selection bias, as the students independently decided their eligibility based on initial screening questions. This voluntary participation may have led to the overrepresentation of students with prior interest or exposure to telerehabilitation, while those with lower awareness or disinterest may have been less likely to participate. Moreover, the study relied on self-reported data, which introduces self-report bias. Students may have overestimated or underestimated their awareness of telerehabilitation due to social desirability or recall bias. Furthermore, differences in the students' confidence levels may have influenced their responses, with some overvaluing their abilities while others undervalued them despite having relevant exposure. Additionally, survey fatigue may have affected response accuracy, as the 42-question survey length could have led to reduced attentiveness or incomplete responses. Although the survey was designed to be concise and self-paced, the possibility of participant fatigue cannot be ruled out.

The use of non-probability sampling in this study presented both strengths and limitations. It facilitated the convenient recruitment of physiotherapy students without logistical limitations, allowing for efficient data collection. Additionally, unequal regional representation may have influenced response, limiting generalizability. Despite these limitations, non-probability sampling was chosen for feasibility and exploratory insights into awareness and application trends. Future studies should use probability-based sampling to enhance representativeness, broaden applicability, and minimize bias. Prior exposure, educational background, or personal interest may influence the student's level of awareness and attitude toward telerehabilitation. Addressing these factors in future research will help to refine the exposure of students to telehealth technologies. Limited contextual factors such as prior exposure to telerehabilitation, clinical experience, or institutional support, could influence awareness levels.

The responses were unevenly distributed, with a dominance of female, 21–25-year-old students, and the undergraduate students, leading to an uneven distribution in gender, academic level, and age. The present study did not explore the variability in education facilities and cultural and socioeconomic factors that influence awareness and perception of telerehabilitation. This study's sample size may not represent the diverse population of physiotherapy students worldwide, which limits the generalizability of the findings. At the time of the study period, no global or domestic database of physiotherapy students was available, which limited direct contact with participants.

## FUTURE DIRECTIONS

Future research should be conducted across diverse regions to explore geographic differences and improve the generalizability of the findings. To integrate telerehabilitation more effectively into sports physiotherapy, we recommend developing standardized curriculum modules focused on key aspects of telerehabilitation, including remote patient monitoring, the use of digital tools for rehabilitation, and the application of artificial intelligence in patient assessment. Future research should adopt longitudinal study designs to track the development of sports telerehabilitation awareness over time, assessing how

exposure, education, and clinical experiences influence knowledge and application. To effectively integrate telerehabilitation into physiotherapy education, policymakers should establish national guidelines that define core competencies in digital health and telerehabilitation. These policies should mandate the inclusion of telerehabilitation training in physiotherapy curricula, ensuring standardized education across institutions. Collaborative efforts with technology providers such as partnerships with developers of e-health platforms and wearable health devices can create comprehensive training that offers students hands-on experience with real-world technologies. Establishing national and international guidelines to standardize sports telerehabilitation practices is crucial along with formulating policies to address data privacy, and ethical frameworks for ensuring the ethical practice of telerehabilitation in the healthcare system.

## ACKNOWLEDGEMENTS

The authors thank the participants who responded to the survey and extend our sincere thanks to those who spread the survey to participants in various parts of the world. The authors would also like to acknowledge Dr. Selvakumar for his valuable assistance in professionally proofreading the manuscript. The authors would like to acknowledge the use of the AI tool, QuillBot, for assistance in grammar correction and paraphrasing during the preparation of this manuscript. Their support in enhancing the clarity and readability of our work is greatly appreciated.

### Funding

The authors received no funding for this work.

### Competing Interests

The authors declare that they have no competing interests.

### Author Contributions

- Vinodhkumar Ramalingam conceived and designed the experiments, performed the experiments, analyzed the data, prepared figures and/or tables, authored or reviewed drafts of the article, and approved the final draft.
- Jeevarathinam Thirumalai conceived and designed the experiments, performed the experiments, analyzed the data, prepared figures and/or tables, authored or reviewed drafts of the article, and approved the final draft.
- Ling Shing Wong conceived and designed the experiments, authored or reviewed drafts of the article, and approved the final draft.
- Rajkumar Krishnan Vasanthi conceived and designed the experiments, authored or reviewed drafts of the article, and approved the final draft.
- Vinosh Kumar Purushothaman performed the experiments, authored or reviewed drafts of the article, and approved the final draft.
- Prathap Suganthirababu analyzed the data, authored or reviewed drafts of the article, and approved the final draft.
- Buvanesh Annadurai conceived and designed the experiments, analyzed the data, prepared figures and/or tables, authored or reviewed drafts of the article, and approved the final draft.

## Human Ethics

The following information was supplied relating to ethical approvals (*i.e.*, approving body and any reference numbers):

The study was formally approved by the Scientific Review Board on Human Subjects of Saveetha College of Physiotherapy, Saveetha Institute of Medical and Technical Sciences (SIMATS) (Approval No. 03/001/2023/ISRB/UGSR/SCPT).

## Data Availability

The raw data is available in the Supplemental File.

## Supplemental Information

Supplemental information for this article can be found online at http://dx.doi.org/10.7717/peerj.19829#supplemental-information.

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
