# Peer review of "Exploring telerehabilitation awareness, application, and future outlook in sports rehabilitation among physiotherapy students: a web-based survey"

_PeerJ, doi:10.7717/peerj.19829_

## Round 0.1 · original submission · Minor Revisions

Please respond in detail to all the comments of the reviewers

·

Basic reporting

The article was clear and well written.Introduction and background were reasonable with the paper premise. Literature references were sufficient. The tables are comprehensive and helpful.

The article brings a good perspective about why telerehabilitation should be in the physical therapist curriculum. The study addresses the objective that is to understand the awareness, application, and future outlook of telerehabilitation in sports physiotherapy among physiotherapy students.

Experimental design

The study is relevant and aligns with the aims and scope of the journal.

The research question is relevant and the description of the methods is good, but there a few suggestions:
1- Detail the inclusion/exclusion criteria
2- More details about the process to deliver the surveys. The e-mails or messages were sent to the students from a database? Universities were contacted?

Validity of the findings

The findings were relevant to propose a modification in the physical therapy curriculum. The data was well presented in the tables.

Just a few adjustments suggested:
1- On table 1, please include the legend for abbreviations. E.g: UG, PG. Ensure all acronyms/abbreviations are defined on first usage in text. If the abbreviation was only used once, do not include the abbreviated form.
2- On line 147 – “Of the 587 students who filled out the survey, 524 were eligible as per the selection criteria”. Please describe why 63 students were excluded. Explain how the study size was arrived at 524 and the exclusion reasons.
3- For the first paragraph of the discussion, would be good to have a resumed finding for the results. Please summarize key results with reference to study objectives.
4- On line 462 – Please rephrase: “Studies have also shown that telerehabilitation will continue to grow and fill the gap associated with Telerehabilitation” and please include a support citation.
5- On line 491- “Students self-selected themselves into the study, which may introduce selection bias.” How they were self-selected?
6- Please expand limitations. Discuss limitations of the study, considering sources of potential bias or imprecision. E.g.: self-report bias.

Reviewer 2 ·

Basic reporting

No coment

Experimental design

No comment

Validity of the findings

Feasibility was noted as an additional aim of this paper, a noted in lines 40 and 41 "feasibility of incorporating sports telerehabilitation into physiotherapy education programs." However, it was assessed by asking if the subjects thought that telerehabiliation is a standard practice in sports rehabilitation. I am not sure this truly assessing how feasible this would be to incorporate this topic into educational programs. Perhaps, this definition of feasibility could be redefined.

Reviewer 3 ·

Basic reporting

Language and Clarity: The manuscript is well-written but not clear. It is written in professional English, with a tone and style appropriate for an academic audience. Technical terms are correctly used, though some sections, especially in the Results and Discussion, could be more concise. Minor grammatical revisions are suggested to improve flow.
Background and Literature References: The introduction provides a solid background and clearly explains the relevance of telerehabilitation in physiotherapy. However, a few recent references could further enhance context, especially given the rapid growth in telehealth adoption since the COVID-19 pandemic. Adding these would strengthen the rationale for investigating student attitudes and awareness.
Structure and Presentation: The article follows a professional structure, with standard sections organised logically. Tables and figures are clear, labelled correctly, and relevant to the presented data.
Self-Contained Study: The manuscript appears self-contained and adequately addresses the stated hypotheses and research objectives. The results are relevant and clearly connected to the research questions, contributing to a cohesive and publishable unit.

Experimental design

Relevance and Contribution to Knowledge: The study focuses on telerehabilitation in sports rehabilitation within physiotherapy education—a timely and relevant topic given the increasing integration of digital tools in healthcare. The research question is well-defined, and the identified knowledge gap is meaningful, emphasizing the need for future practitioners to be skilled in telerehabilitation.
Methodological Rigor: The methods section is detailed, but clarifying the selection criteria for participants could improve understanding of the study's scope. Additionally, describing survey content and scales in greater depth would aid replicability, as would elaborating on any limitations related to sample representativeness. The study adheres to the ethical standards mentioned in the manuscript, but adding an ethics committee reference would confirm this.
Reproducibility: Most methods are described with adequate detail.

Validity of the findings

Robustness of Data and Statistical Soundness: The data analysis is statistically sound, using appropriate p-values. However, detailing any potential confounding factors (e.g., prior exposure to telehealth technologies) could strengthen the robustness of the findings.
Conclusions and Research Relevance: Conclusions are well-stated and directly linked to the original research question, summarizing key insights on physiotherapy students' awareness and potential for using telerehabilitation. The limitations section could be expanded to address generalizability, especially given the study's sample, which may not fully represent all physiotherapy students globally.
Value to Literature: This study contributes to a growing body of knowledge on telerehabilitation and its future in healthcare education. The findings provide valuable insights into the attitudes of emerging physiotherapists and offer guidance for incorporating telehealth into training. The study fills a relevant knowledge gap, particularly for educational institutions considering curriculum updates in response to technological advancements.

Additional comments

Abstract: Include key findings to make the abstract more impactful.
Literature and Background: To strengthen the rationale for the study, consider adding recent statistics and references on the growth of telerehabilitation, particularly in sports medicine.
Methods: Specify inclusion and exclusion criteria and add information on sample representativeness.
Discussion: Discuss potential biases, like prior exposure to telehealth, and elaborate on how findings could inform curriculum updates in telerehabilitation training to enhance the study’s practical impact.
References: Ensure that all are formatted consistently and check for any recent studies that might add value, particularly those that have emerged in response to the increased use of telehealth during the COVID-19 pandemic.
General comments: Some sentences could be made more concise for easier reading, particularly in the results and discussion sections.

Reviewer 4 ·

Basic reporting

Including more recent references on telerehabilitation adoption trends, especially in sports contexts, would provide a stronger foundation for the study's relevance.

Additional context on how telerehabilitation specifically benefits sports rehabilitation, compared to other fields, would strengthen the argument for its inclusion in physiotherapy curriculums.

More clarity on how questions were formulated and validated would improve the rigor of the study. If the questionnaire was piloted or based on previous studies, please mention this to enhance the study's methodological transparency.

Provide more information about the sample demographics, such as age ranges, educational institutions, and geographic distribution. Explaining the participant selection process and response rate would improve the study's generalizability.

could be strengthened by linking specific results to broader implications for physiotherapy education and policy. Discussing limitations, such as the potential for response bias or the study's cross-sectional design, would provide a more balanced view.

Future research directions, especially on implementing telerehabilitation training in physiotherapy curriculums, would be valuable additions.

Experimental design

The study on telerehabilitation awareness and perspectives among physiotherapy students fits well within the journal’s focus on rehabilitation sciences and healthcare technology, making it relevant to readers interested in emerging trends in physiotherapy and digital health solutions.

The research question is clearly stated and relevant, addressing an important area in healthcare by exploring how future physiotherapists perceive telerehabilitation. The authors articulate a knowledge gap in the understanding of telerehabilitation readiness in physiotherapy education, especially with sports applications, making the study both meaningful and timely.

Details on any ethical approval obtained, or the informed consent process for participants, would improve clarity on compliance with ethical guidelines.

The methods are described sufficiently for replication, with a clear structure of the questionnaire provided. Expanding on how the survey was distributed and clarifying participant selection criteria would further enhance replicability.

Validity of the findings

While the study addresses an important topic—awareness and readiness for telerehabilitation among physiotherapy students—it would benefit from a more explicit statement on its impact on future physiotherapy training and healthcare practices. Highlighting the novelty in terms of how this research can influence educational policies or integration of telerehabilitation in curriculums would strengthen its contribution.

The study has potential for meaningful replication across different populations and educational institutions, given the increasing role of telerehabilitation. Stating the benefit of replicating this study in diverse geographic and institutional contexts would add value to the literature and encourage future research.

The data provided in the questionnaire format is comprehensive, covering awareness, application, and future outlook. However, including raw statistical data summaries (e.g., frequencies, percentages) in tables or figures would make the findings more accessible and validate the robustness of the study’s data.

The conclusions are generally well-stated, with clear links to the original research question on telerehabilitation awareness and perspectives. Expanding on how the findings directly address the identified knowledge gap and stating the practical applications for physiotherapy education would further strengthen this section.

---

## Round 0.2 · Major Revisions

Please respond in detail, particularly to the comments from Reviewer 4

·

Basic reporting

no comment

Experimental design

no comment

Validity of the findings

no comment

Additional comments

Everything that I suggested was addressed and I have no other comments.

Reviewer 2 ·

Basic reporting

Overall with acceptable changes, one note is that the tense use is not consistent. In line 236: "The maximum score for application knowledge part is 65 (13 items)." The past tense use of "was" instead of "is" should be used.

In line 486 telerehabilitation technologies. "Physiotherapy students show a moderate level of awareness and" past tense used would be "showed"

A lot of the results section was written in present tense.

Experimental design

No comment

Validity of the findings

No comment

Reviewer 3 ·

Basic reporting

While the article is written in generally clear English, occasional awkward phrasings could benefit from professional editing. For example, sentences discussing the integration of telerehabilitation into curricula lack conciseness and clarity. Revising these areas will improve readability and comprehension.

Professional article structure, figures, tables, and raw data shared:
The article follows a professional structure. However, the quality of some figures and tables could be enhanced to improve readability.
Ensure all figures have consistent formatting and captions clearly explaining the data.
Provide more detailed labels in tables to assist readers in interpreting results without extensive cross-referencing.

The Discussion section could highlight the broader implications of results, particularly in overcoming barriers to integrating telerehabilitation into education and practice.

Experimental design

The study is original and falls within the journal’s aims and scope. However, the novelty of the work could be better highlighted. Specifically, discussing how this research uniquely contributes to addressing knowledge gaps in telerehabilitation education and implementation would strengthen the manuscript.

Clarify why these 42 survey items were chosen and how they align with the knowledge gaps identified in previous literature.

Expand on how regional and demographic variations (e.g., gender differences) provide actionable insights for telerehabilitation adoption.

The authors should clarify whether the sample is representative of the population and how selection bias, if any, was mitigated.
The rationale for excluding any survey responses, if applicable.

The Methods section is thorough but could benefit from more detail in the following areas:
Explain the reasoning behind the structure and content of the survey.
Provide more specifics on how data were analyzed, including details on statistical tests and their appropriateness for the data type.
Add information on how missing/incomplete data were handled.

Validity of the findings

While the findings are meaningful and relevant, the manuscript could strengthen its discussion of how these results encourage replication. For instance:
Provide examples of how this study’s methodology could be adapted to other regions or demographics.
Discuss how similar research in other healthcare sectors could validate the findings.

Conclusions are well stated, linked to the original research question & limited to supporting results:
The conclusions are well-stated but could be expanded to include actionable recommendations. For example:
Based on the findings, suggest specific frameworks or models for integrating telerehabilitation into curricula.
Highlight how the study’s insights could influence policy or practice.

Additional comments

The Discussion section could better address the practical implications of the findings, particularly those related to technological challenges such as internet connectivity and resource constraints.

Ensure consistent formatting across the manuscript, particularly in referencing and citation style.

Reviewer 4 ·

Basic reporting

Basic Reporting
1. Language and Clarity:
• The manuscript is generally well-written, but there are instances of repetitive phrasing and overly complex sentence structures. For example, the abstract and introduction could be simplified for clarity and conciseness.
• Suggested Revision: Revise the introduction and abstract to streamline ideas and avoid redundancy. Use professional proofreading to improve readability.
2. Background and Literature Review:
• The introduction provides sufficient context for the study but lacks integration of some key recent studies in telerehabilitation. For example, advancements in wearable technology and AI-enhanced telerehabilitation are only briefly mentioned.
• Suggested Revision: Incorporate more comprehensive references to recent advancements in digital health technologies to strengthen the foundation of the study.
3. Structure and Figures:
• The structure adheres to academic standards, but some figures and tables lack descriptive captions, reducing their standalone interpretability (e.g., Tables 1 and 3).
• Suggested Revision: Enhance figure and table captions to explain their content fully. For instance, clearly define terms like “quartile levels” in Table 3.
4. Data Sharing:
• Raw data is available, but additional metadata or explanatory notes about the dataset would make it more accessible for replication.
• Suggested Revision: Provide clear documentation alongside the raw data to facilitate reproducibility.

Experimental design

1. Research Question:
• The study addresses a meaningful knowledge gap concerning the awareness, application, and future outlook of telerehabilitation among physiotherapy students. The research question is well-defined.
2. Methodology:
• Strengths: The use of a validated questionnaire and adherence to the CROSS guidelines are commendable.
• Weaknesses: The sampling strategy (non-probability sampling) limits generalizability. Additionally, the exclusion criteria and rationale for categorization in demographic analyses require further explanation.
• Suggested Revision:
• Justify the use of non-probability sampling and discuss its implications.
• Elaborate on the exclusion criteria (e.g., lack of English proficiency) and their potential impact on study outcomes.
3. Ethical Standards:
• The study meets ethical standards, with Institutional Review Board approval and anonymization of responses clearly stated.
4. Methodological Clarity:
• The methods section includes sufficient detail for replication, but some technical aspects, such as the justification for statistical tests (e.g., multinomial logistic regression), could be better explained.
• Suggested Revision: Include a brief explanation of why specific statistical models were used.

Validity of the findings

1. Data and Statistical Analysis:
• Strengths: The data is statistically sound, with appropriate reliability testing (e.g., Cronbach’s alpha) and significance testing (e.g., chi-square, logistic regression).
• Weaknesses: Some results, such as demographic influences on future outlook, lack depth in interpretation.
• Suggested Revision: Expand on the implications of key findings, particularly demographic predictors, to provide actionable insights. For instance, why do younger students show less application knowledge?
2. Conclusions:
• The conclusions align with the data but tend to overstate the readiness of physiotherapy students to adopt telerehabilitation as standard practice. For example, the moderate levels of awareness and application knowledge suggest more barriers than the conclusion implies.
• Suggested Revision: Reframe the conclusions to reflect the mixed findings, emphasizing the need for educational reforms to improve awareness and application skills.
3. Comparative Analysis with Literature:
• The manuscript references relevant studies but does not fully compare its findings to the existing body of literature. For example, the discussion could better highlight how the study’s results align or contrast with previous research in related fields.
• Suggested Revision: Include a comparative analysis of findings with those of similar studies to contextualize the results more effectively.

Additional comments

While the manuscript contributes valuable insights into telerehabilitation awareness and application, it requires substantial revisions to enhance clarity, methodological transparency, and contextual integration within existing literature. Addressing these issues will significantly improve the study’s impact and relevance.

1. Limitations Section:
Expand the limitations section to address the potential biases introduced by self-reported data and non-probability sampling. Discuss how these limitations could be mitigated in future studies.
2. Future Directions:
Provide more specific recommendations for integrating telerehabilitation into curricula, such as example modules or collaborative efforts with technology developers.
3. Figures and Tables:
Improve the clarity of all visual aids, ensuring they are fully interpretable without referencing the main text.

---

## Round 0.3 · Minor Revisions

Your submission still needs some remaining revisions. Please respond to the comments below

Reviewer 3 ·

Basic reporting

Clarity & Language:
The manuscript is well-structured but contains grammatical errors and awkward phrasing. Consider professional proofreading to improve readability.
Example: The sentence "Physiotherapy students showed a moderate level of awareness and a moderate to high level of application knowledge towards telerehabilitation." could be restructured for better flow.
Avoid redundancy (e.g., "moderate level of awareness" appears multiple times without deeper insights).

Literature & Context:
The introduction is well-researched but could benefit from a more precise discussion of existing gaps in telerehabilitation studies.
Some references appear outdated or lack direct relevance to the study. Consider integrating recent systematic reviews or meta-analyses to strengthen the literature base.

Figures & Tables:
Figures are clear, but some lack detailed captions.
Table 3 does not clearly define "moderate" and "high" awareness levels—explicitly state the cut-off values.

Experimental design

Research Question & Rationale:
The study aims to explore awareness and application, but the research question could be framed more explicitly.
The rationale behind using a non-probability sampling method is not sufficiently justified. Discuss limitations related to sampling bias.

Methodology:
The questionnaire validation process is unclear. How was validity determined? Mention if any pilot testing was done.
How were students recruited across different regions? If online recruitment was used, mention if regional disparities in internet access may have affected participation.
The study lacks details on how the survey was standardized across different cultural and educational backgrounds.

Validity of the findings

Data Presentation:
The study provides rich data, but some statistical results (Chi-square tests and logistic regression) need effect sizes to indicate the practical significance.
The discussion on gender and regional disparities is interesting but lacks clear explanations for observed differences.
The relationship between demographic variables and awareness is not always well-explained (e.g., why are younger students more aware but have lower application knowledge?).

Statistical Analysis:
The study does not clearly define awareness levels (e.g., how were quartiles determined?).
The ordinal logistic regression model results should be interpreted more clearly—highlight the key predictors and their implications for education policy.
Results should be reported with confidence intervals where appropriate.

Additional comments

Limitations:
While limitations are acknowledged, they should include response bias (students self-reporting knowledge may overestimate their awareness).
The study does not mention whether survey fatigue was considered—was the 42-question survey too long for participants to answer accurately?

Future Directions:
Suggest longitudinal studies to track awareness development over time.
Provide recommendations for how telerehabilitation could be incorporated into physiotherapy curricula.

Reviewer 4 ·

Basic reporting

1. Basic Reporting

Language Clarity & Professional English
• Issue: The manuscript is generally well-written but contains grammatical inconsistencies and awkward phrasing in several sections. For example, in the abstract and introduction, there are redundant phrases that could be streamlined for better readability.
• Recommendation: A thorough language edit is necessary to ensure clarity. For instance, the sentence “Our study aims to evaluate the level of awareness, the practical implementation and the future perspectives of physiotherapy students regarding the sports telerehabilitation.” can be revised to “This study evaluates physiotherapy students’ awareness, implementation, and perspectives on sports telerehabilitation.”

Literature References & Background
• Issue: The introduction provides historical context on sports physiotherapy but does not sufficiently connect it to recent advancements in telerehabilitation. Several key studies in the last five years are missing.
• Recommendation: Incorporate more recent literature (2020–2024) on the effectiveness of telerehabilitation in sports physiotherapy, particularly regarding its integration into educational curricula.

Article Structure & Figures/Tables
• Issue: The article follows an appropriate structure but has inconsistent formatting in table descriptions. Some figure captions lack enough explanation for standalone interpretation.
• Recommendation: Improve table legends to clarify statistical outcomes and data interpretation, particularly in Tables 3 and 5.

Raw Data Availability
• Issue: The raw data is provided, which is commendable. However, additional metadata or a clear description of variables would enhance transparency.
• Recommendation: Include a supplementary document explaining the raw data structure for reproducibility.

Experimental design

Research Question & Relevance
• Issue: The research question is relevant but could be more explicitly connected to a specific knowledge gap in the field.
• Recommendation: Clearly state the unique contribution of this study compared to existing surveys on telerehabilitation awareness among students.

Methodology Transparency
• Issue: While the methodology is generally robust, the study does not justify the exclusion of students without prior telerehabilitation exposure.
• Recommendation: Explain why students with minimal knowledge of telerehabilitation were allowed to continue the survey. Additionally, clarify the validity and reliability testing process for the questionnaire.

Ethical Considerations
• Issue: Ethical approval is mentioned but lacks details on participant consent.
• Recommendation: Specify whether informed consent was explicitly obtained from all participants.

Validity of the findings

Data Robustness & Statistical Soundness
• Issue: The study employs appropriate statistical techniques (chi-square tests and regression models), but the interpretation of some findings lacks depth.
• Recommendation: Discuss potential confounders that could influence awareness and application levels. For example, prior exposure to telerehabilitation outside formal education could skew results.

Conclusions & Research Question Alignment
• Issue: The conclusions align with the results but could benefit from a more nuanced discussion on how findings compare with prior studies.
• Recommendation: Consider discussing why younger students had higher awareness but lower application proficiency, referencing digital literacy frameworks.

Additional comments

• The discussion section could be more concise, as certain arguments are repeated (e.g., benefits of telerehabilitation are reiterated in multiple places).
• Consider adding recommendations for policymakers on integrating telerehabilitation into physiotherapy curricula.
• Expand on limitations regarding self-reported data and potential selection bias.

1. Revise language and grammar for clarity and conciseness.
2. Expand literature review to include recent studies on telerehabilitation adoption in educational settings.
3. Clarify methodology, particularly regarding the justification for participant selection criteria.
4. Improve statistical discussion by addressing potential confounders and alternative interpretations of findings.
5. Enhance conclusion section by providing more concrete policy and curriculum recommendations.

---

## Round 0.4 · Minor Revisions

Your work can substantially contribute to the field, but you would benefit from being more succinct and having a full review for English language editing. your statistical methods are appropriate, but some interpretations of tests require clarification (See reviewer comments).

**Language Note:** The review process has identified that the English language must be improved. PeerJ can provide language editing services - please contact us at [email protected] for pricing (be sure to provide your manuscript number and title). Alternatively, you should make your own arrangements to improve the language quality and provide details in your response letter. – PeerJ Staff

Reviewer 3 ·

Basic reporting

Abstract (Lines 37–55):
Consider breaking long sentences for readability.
Example: “A validated, self-designed questionnaire…” → split this sentence into two.

Introduction:
Some sentences are overly descriptive or redundant.
Example (Line 73): "Telerehabilitation had a widespread impact..." → revise to “Telerehabilitation has had a widespread impact…”
Syntax Note: Replace awkward phrases like "stood in the student phase as 50.3%" with clearer alternatives, such as "50.3% of students demonstrated moderate awareness."
L121: Incorrect reference.
L160-163: Should be corrected to: "Of the 587 students who completed the survey, 524 were eligible according to the selection criteria. A total of 63 students were excluded due to not completing their sports physiotherapy module at the time of the survey (n=42), duplicate responses (n=14), or not having a background in physiotherapy (n=7)."
Line 177–180: The Ethics sentence can be improved to: "Ethical approval was obtained from the Institutional Scientific Review Board of Saveetha College of Physiotherapy, SIMATS (Approval No. 03/001/2023/ISRB/UGSR/SCPT)."

Results
L278-180: A sentence cannot be a paragraph.
Clarify some regression result interpretations — some explanations are difficult to follow.
Line 301–309: Phrase "stands in the student phase as 36.5%" could be rephrased to: “36.5% of students reported a moderate level of application knowledge…”

General Comments
Clarity & Readability: The manuscript would benefit from a careful language edit for smoother syntax and grammar. Avoid repetitive phrases like "strongly agreed or agreed" — consider condensing when appropriate.

Terminology: Use consistent terms for categories, e.g., "moderate awareness" instead of “student phase of awareness.”

Figures/Tables: Make sure supplemental tables referenced (e.g., Table S5, S6) are clear and available for peer reviewers.

Experimental design

-

Validity of the findings

-

Reviewer 4 ·

Basic reporting

No comments

Experimental design

No comments

Validity of the findings

No comments

---

## Round 0.5 · Minor Revisions

Thank you for your efforts in addressing the previous feedback. Minor amendments to the consistency of statistical reporting are needed before your manuscript can be accepted for publication. Please see the reviewer comments and provide a response to their comments as you see fit.

**Language Note:** The review process has identified that the English language must be improved. PeerJ can provide language editing services - please contact us at [email protected] for pricing (be sure to provide your manuscript number and title). Alternatively, you should make your own arrangements to improve the language quality and provide details in your response letter. – PeerJ Staff

Reviewer 3 ·

Basic reporting

See Additional comments.

Experimental design

See Additional comments.

Validity of the findings

See Additional comments.

Additional comments

Language and Grammar
The Introduction and Discussion are overly verbose and repetitive.
Sentences such as "Students perceived a benefit in utilizing telerehabilitation for personalized AI-guided routines..." could be tightened to improve clarity and flow.
Consider a full proofreading pass for stylistic polish and conciseness.

Clarity in Survey Design
More detail is needed on how quartiles were established and how the scoring system for awareness, application, and outlook was validated. Consider adding this to the Methods or Supplemental Material.

Statistical Reporting
Although regression results are included, ensure that effect sizes, CIs, and p-values are reported consistently.
Include a brief interpretation of ORs in lay terms (e.g., “twice as likely” instead of “OR = 2.0”) where appropriate for broader reader comprehension.

Abstract Optimisation
Consider refining the abstract to remove redundancy. Keep focus on key outcomes and implications. It’s currently too detailed for journal abstracts.

---

## Round 0.6 · Minor Revisions

Thank you for your revised manuscript and attention to the changes suggested by the reviewers.

Before the manuscript can progress, some further adjustments to the language and interpretation are required.

1. You have amended the manuscript to use the term "tertile". In this instance, the use of tertiles is inappropriate because your divisions do not create three equal groups separated by response/value frequency. Your divisions are representative of quartiles, but you have combined the first and second quartiles into one group.

Please address this amendment and any corresponding areas of the manuscript to accurately reflect the values that are reported and the methods used to generate these data.

---

## Round 0.7 · accepted · Accept

Thank you for addressing the comments of the reviewers. I am pleased to advise that you manuscript is now ready for acceptance.

Reviewer 3 ·

Basic reporting

.

Experimental design

.

Validity of the findings

.

Additional comments

.